# RYBP Sensitizes Cancer Cells to PARP Inhibitors by Regulating ATM Activity

**DOI:** 10.3390/ijms231911764

**Published:** 2022-10-04

**Authors:** Deanna V. Maybee, Alexandra Maria Psaras, Tracy A. Brooks, Mohammad A. M. Ali

**Affiliations:** Department of Pharmaceutical Sciences, SUNY Binghamton University School of Pharmacy and Pharmaceutical Sciences, Binghamton, NY 13790, USA

**Keywords:** Ring1 and YY1 Binding Protein (RYBP), ataxia telangiectasia mutated (ATM), DNA damage response (DDR), checkpoint kinase 2 (Chk2), poly-ADP-ribose polymerase (PARP)

## Abstract

Ring1 and YY1 Binding Protein (RYBP) is a member of the non-canonical polycomb repressive complex 1 (PRC1), and like other PRC1 members, it is best described as a transcriptional regulator. Previously, we showed that RYBP, along with other PRC1 members, is also involved in the DNA damage response. RYBP inhibits recruitment of breast cancer gene 1(BRCA1) complex to DNA damage sites through its binding to K63-linked ubiquitin chains. In addition, ataxia telangiectasia mutated (ATM) kinase serves as an important sensor kinase in early stages of DNA damage response. Here, we report that overexpression of RYBP results in inhibition in both ATM activity and recruitment to DNA damage sites. Cells expressing RYBP show less phosphorylation of the ATM substrate, Chk2, after DNA damage. Due to its ability to inhibit ATM activity, we find that RYBP sensitizes cancer cells to poly-ADP-ribose polymerase (PARP) inhibitors. Although we find a synergistic effect between PARP inhibitor and ATM inhibitor in cancer cells, this synergy is lost in cells expressing RYBP. We also show that overexpression of RYBP hinders cancer cell migration through, at least in part, ATM inhibition. We provide new mechanism(s) by which RYBP expression may sensitize cancer cells to DNA damaging agents and inhibits cancer metastasis.

## 1. Introduction

When DNA damage occurs inside a cell, a response pathway is activated, known collectively as the DNA damage response (DDR) [1]. DDR pathways, in general, aim to halt the cell cycle and repair the damage [1]. The cellular response is dependent on the type of DNA damage present. The list of the DDR proteins that respond to double strand breaks (DSB), one of the most deleterious types of DNA damage [1], is growing and includes the RING1 and YY1 Binding Protein (RYBP) [2]. Other bona fide DDR proteins include the protein kinase ataxia telangiectasia mutated (ATM) [3] and the enzyme poly-ADP-ribose polymerase (PARP) [4]. Each DDR protein contributes to the repair process, yet they respond to DNA damage in various ways according to the type and stage of DNA damage [5].

RYBP is a multifunctional protein, serving as a ubiquitin-binding protein that preferentially binds to K63-linked ubiquitin chains [2], RING1B, histone H2A, and other transcription factors [6,7]. We specifically examined RYBP’s role in DDR and revealed that RYBP is actively removed from DNA damage sites due to its ability to bind to K63-ubiquitin chains at sites of double strand breaks (DSBs) [2]. As a result of RYBP’s ubiquitin binding abilities, RYBP impairs breast cancer gene 1 (BRCA1) complex recruitment and inhibits homologous recombination repair [2]. RYBP is also shown to effectively compact the chromatin [8], an effect that is less favorable at DSB sites. Thus, RYBP overexpression sensitizes cancer cells to DNA damaging agents [2,9].

ATM is a protein kinase that is early activated in response to DNA damage [4,10,11,12]. Initially, ATM is present as an inactive dimer and later monomerizes in the presence of DSBs [13]. ATM activation results in a cascade of phosphorylation reactions beginning with its autophosphorylation and leading to the phosphorylation of numerous downstream targets, including cell cycle checkpoint 2 (Chk2) [13,14,15]. The phosphorylated ATM decondenses chromatin at DNA damage sites to increase the accessibility for DNA repair proteins, while phosphorylated Chk2 enforces S- and G2-M cell cycle arrest [3,10]. Contrary to RYBP, inhibiting ATM activity in cancer cells sensitizes them to DNA damage [16].

ATM in cancer cells tend to be upregulated more often than downregulated [3]. One possible explanation for this is that the activation of the oncogene in precancerous stages strains DNA replication. This may result in the activation of DDR and leads to persisted ATM signaling in malignant tumors, albeit in cancer cells ATM is no longer interconnected to cell-cycle arrest and apoptosis [3]. When ATM is inhibited or inactivated in cancer cells (including colorectal, pancreatic, lung, breast, and gastric cancers) [17], they become radiation sensitive due to defects in DNA damage repair [4]. As a result of these findings, a few clinical trials using ATM inhibitors—AZD0156, KU-60019 and AZD1390—were conducted for solid tumors [17]. These phase-I trials sought to determine the tolerable dose of ATM inhibitors and their effects when combined with radiation therapy or additional inhibitors, such as the PARP inhibitors; olaparib and pazopanib. Some of these clinical trials are still recruiting patients and the results are pending (ClinicalTrials.gov Identifiers: NCT02588105, NCT03571438, NCT03423628).

PARP is a crucial enzyme in base excision repair and single strand break (SSB) repair pathways, as it binds to ends of DNA at both SSB and DSB sites for repair [18,19]. When PARP is inhibited, this blocks the repair of DNA strand breaks and sensitizes cancer cells to various DNA damaging agents [4]. A loss of PARP results in an increase in the number of DNA damages repaired by homologous recombination, and thus PARP inhibitors are particularly efficacious in the tumor context of inhibited homologous recombination DNA repair mechanisms, such as BRCA mutations [20,21]. Olaparib, the first FDA-approved PARP inhibitor [22], reduces cell proliferation by inducing replication stress [4]. Weston et al. (2010) examined olaparib efficacy separate from in BRCA deficient cancers to determine the effect it had on ATM-deficient lymphoid tumor cells [20]. It was reported that ATM deficient lymphoid cells had a greater sensitivity to PARP inhibitors [20]. A similar sensitivity of olaparib in ATM deficient cells was also observed in two breast cancer cell lines [23].

Therefore, each component of the DDR plays an important role in DNA repair pathways, and they are ideal targets in cancer therapy. In our previous studies [2], we reported that high levels of RYBP tended to reduce the phosphorylation of H2A.X (an ATM substrate). In this study, we hypothesize that RYBP inhibits ATM activity in the presence of DNA damage. ATM inhibition by RYBP may further provide a new mechanism by which RYBP sensitizes cancer cells to DNA damage and PARP inhibitors. We report here that overexpression of RYBP inhibits phosphorylation and recruitment of ATM to DNA damage sites. RYBP is also shown to inhibit phosphorylation of Chk2 upon DNA damage. Last but not least, we found that cancer cells that overexpress RYBP show higher sensitivity to PARP inhibition and lower migration. These effects were found to be mediated, at least in part, by ATM inhibition.

## 2. Results

### 2.1. RYBP Inhibits ATM Activity and Autophosphorylation Induced by DNA Damage

To investigate whether RYBP inhibits ATM activity, GFP-RYBP expressing U2OS cells were treated with camptothecin or calicheamicin and immunostained for RYBP and phosphorylated ATM (p-ATM). Phosphorylation of ATM is indicative of ATM activity, as autophosphorylation and recruitment to DSB sites occurs at early stages of DNA damage [24]. Immunofluorescence revealed that control cells exhibited several p-ATM foci inside the nucleus upon DNA damage. On the other hand, cells expressing RYBP had fewer p-ATM foci when treated with either calicheamicin or camptothecin (Figure 1A,B). These results suggest that expression of RYBP inhibits ATM phosphorylation and recruitment to DNA damage sites.

Although calicheamicin causes DNA damage similar to radiation (radiomimetic) and camptothecin causes DSB during S-phase in cells replicating DNA [25], the number of p-ATM foci in GFP-RYBP cells were comparable in both treatments (Figure 1A,B). When p-ATM foci were analyzed and averaged, there was a significant difference in the number of foci in control cells compared to the GFP-RYBP expressing cells.

Calicheamicin-treated cells averaged 23 p-ATM foci per control cell and averaged 4 p-ATM foci per GFP-RYBP cell, while camptothecin-treated cells averaged 24 ATM foci per control cells and 3 ATM foci per GFP-RYBP expressing cells (Figure 1, right panels). These findings indicate that calicheamicin- and camptothecin- induced damage results in similar ATM activation, and there is a significant reduction in ATM phosphorylation/activity in RYBP expressing cells compared to control cells. Consequently, this suggests that RYBP reduces or inhibits ATM activity and recruitment to DNA damage sites.

### 2.2. RYBP Inhibits Phosphorylation of Chk2 upon DNA-Damage

To further support that RYBP inhibits ATM activity, the effect of RYBP on the phosphorylation of Chk2 (p-Chk2), a downstream substrate of ATM, was determined. U2OS cells expressing GFP (control) and those expressing GFP-RYBP were treated with 1 µM camptothecin for 1, 6, and 10 h to induce DNA damage [26]. Those cells at 0 time represent untreated controls.

Western blots showed that GFP cells showed progressively increased levels of p-Chk2 at 1-, 6-, and 10-h time intervals, while GFP-RYBP cells only reflected a slight increase in p-Chk2 1 h after camptothecin treatment (Figure 2A). A decrease in p-Chk2 levels, in both the GFP control and GFP-RYBP, occurred at 10 h after camptothecin treatments. After calculating the p-ChK2 to total Chk2 ratio, a clear trend exists across camptothecin treatments (Figure 2B). A distinct difference in levels of p-Chk2/Chk2 ratio is illustrated between GFP and GFP-RYBP at 6 h and 10 h after camptothecin treatments (Figure 2B). The high p-Chk2/Chk2 ratios are directly indicating ATM activity [1]. In GFP-controls, the p-Chk2/Chk2 ratio was significantly increased when DNA damage was induced at 6- and 10-h time intervals, correlating to higher ATM activity.

On the other hand, the lack of significant changes in the p-Chk2/Chk2 ratios in cells expressing GFP-RYBP reinforces our findings that high levels of RYBP inhibit the phosphorylation of Chk2 (an ATM substrate), and thus, ATM activity upon DNA damage.

To further support our findings that overexpressing RYBP reduces the phosphorylation of Chk2, we also repeated these experiments in ovarian cancer cell line, SKOV3. There were distinct reductions in pChk2/Chk2 ratios at 1 h and 10 h after camptothecin treatments in GFP-RYBP expressing cells (Appendix A). Although the trend is clear, these differences did not reach statistical significance in SKOV3 (one-way ANOVA) as in U2OS. The results, however, suggest a potential role of RYBP in reducing ATM activity, and thus phosphorylation of Chk2, in two different cancer cell lines.

### 2.3. RYBP Sensitizes Cancer Cells to PARP Inhibitor by Reducing ATM Activity

To ascertain the effect of RYBP activity on the efficiency of PARP inhibitors, an MTT assay was performed on GFP and GFP-RYBP expressing cells. Cell viability was measured for U2OS cells treated with the following PARP inhibitor (ABT-888) concentrations: 2.5, 5.0, 10, 20, 50 and 100 µM. As the PARP inhibitor concentration increased, the percentage of surviving cells gradually declined in both GFP and GFP-RYBP cells (Figure 3A,B). Although both groups showed a decline, there was an increase in effectiveness PARP inhibitor had on RYBP-expressing cells compared to the GFP control cells. This is evident after calculating the IC_50_ of the PARP inhibitor for the GFP control and RYBP-expressing cells; 220 µM vs. 83 µM, respectively. These results show that RYBP sensitizes U2OS cells to PARP inhibitor.

To establish whether the cause of this sensitivity to PARP inhibitor is a result of RYBP inhibiting ATM activity, the MTT assay was also conducted on GFP control and GFP-RYBP cells in the presence of the ATM inhibitor, KU-55933. Cells (Control and RYBP-expressing) were treated with a combination of PARP inhibitor (increasing concentrations as previously listed) and 5 µM of ATM inhibitor (Figure 3A,B). When cell viability was measured, there was a similar result in RYBP-expressing cells, as there was no statistical difference between the percentage of surviving cells solely treated with PARP inhibitor and those with the combination of inhibitors (Figure 3B). However, GFP control cells treated with this combination of inhibitors exhibited different results than those observed with RYBP-expressing cells (Figure 3A). By comparing the cell viability between GFP control cells treated with PARP inhibitor or PARP inhibitor and ATM inhibitor (Figure 3A), we observed a significant reduction in cell survival at 20 µM and 100 µM. The IC_50_ in GFP control cells exhibited an 8.5-fold change when comparing the IC_50_ for PARP inhibitor in the absence (=220 µM) or presence (=26 µM) of the ATM inhibitor. These results contrasted those of RYBP-expressing cells, where the IC_50_ for PARP inhibitor in the presence of ATM inhibitor was 53 µM, resulted in less than a 2-fold change from the IC_50_ in the absence of ATM inhibitor (83 µM) (Figure 3B). These results indicate that the combination of PARP inhibitor and ATM inhibitor revealed synergistic effects only in control cells not overexpressing RYBP. In fact, this synergistic effect was lost in cells expressing RYBP and suggests that RYBP sensitizes cancer cells to PARP inhibitor, at least in part, by reducing ATM activity.

Next, we sought to test this effect on the PARP inhibitor-resistant cell line, SKOV3 [27]. Cell viability was measured for the same PARP inhibitor (ABT-888) but we used higher concentrations range: 30, 50, 100, 300, 500, 1000 µM. Our data showed that SKOV3 was less susceptible to the killing effect of the PARP inhibitor, ABT-888, with IC_50_ of 340 µM (Appendix A). Likewise, addition of ATM inhibitor in control cells lowered the PARP inhibitor IC_50_ from 340 µM to 161 µM (Appendix A). On the other hand, Appendix A shows that RYBP-expressing SKOV3 cells showed slightly higher sensitivity to PARP inhibitor (IC_50_ = 269 µM) when compared to control cells (IC_50_ = 340 µM). However, when the ATM inhibitor is combined with PARP inhibitor in RYBP-expressing SKOV3, the IC_50_ of the PARP inhibitor almost did not change (IC_50_ = 235 µM). These results further confirm that the addition of ATM inhibitor in RYBP-expressing cells did not affect the IC_50_ of PARP inhibitor (Appendix A). Even though these results are not as distinct in the PARP-resistant SKOV3 cells as shown in U2OS cells, it further suggests the effect of RYBP on reducing ATM activity.

### 2.4. RYBP Reduces Migration of Cancer Cells via Regulating ATM Activity

Recent findings suggest that ATM inhibition reduces cancer cell migration [28]. To determine the effect RYBP has on cancer cell migration in U2OS (Figure 4) and MDA-MB-231 (Appendix A), a wound closure assay was performed. Non-transfected U2OS control cells were examined and imaged for the duration of 72 h until complete wound closure was observed. Additionally, those cells transfected with RYBP were observed and imaged, showing minimal wound closure at 72 h (Figure 4). The lack of wound closure is seen in cells overexpressing RYBP (Figure 4). These results signify that when overexpressed, RYBP impedes migration of cancer cells.

To establish whether RYBP’s ability to reduce ATM activity is the mechanism responsible for inhibiting cancer cell migration, non-transfected and RYBP-expressing cells were treated with 30 µM of the ATM inhibitor. The wounds, congruous to that of cells expressing RYBP, showed incomplete wound closure and minimal cell migration (Figure 4). These significant differences in wound closure between the non-transfected control, the cells overexpressing RYBP, and the cells with ATM inhibitor, definitively show that both RYBP overexpression and ATM inhibition reduce cell migration to similar extent. However, in cells expressing RYBP, the addition of ATM inhibitor did not significantly reduce cell migration, revealing that hindered cell migration by RYBP can be attributed to reduced ATM activation.

## 3. Discussion

The ATM kinase is involved in cell-cycle arrest, apoptosis, and DDR through the homologous recombination repair pathway [5]. When the DNA is damaged, ATM autophosphorylation occurs on serine 1981 residue, resulting in transition of the inactive ATM dimer to a catalytically active monomers [29]. Therefore, autophosphorylation of ATM after DSBs results in ATM monomerization; the first step in ATM’s activation and recruitment to DNA damage sites in the repair pathway [30]. By its turn, ATM phosphorylates the heterochromatin-building factor KAP-1 to promote the repair of DSBs within heterochromatin regions [5]. Additionally, ATM is responsible for the phosphorylation of various downstream substrates involved in cell cycle checkpoint control and DDR responses, including Chk2 [5,30].

Preclinical and clinical observations demonstrated that disrupting the homologous recombination pathway, by inhibiting ATM, sensitizes cancer cells to various therapies [5]. It was also found that ATM defective lung adenocarcinoma is sensitive to cisplatin and olaparib, as a result of incompetent homologous recombination repair [5]. Perkhofer et al. (2017) also revealed that ATM deletion causes pancreatic ductal adenocarcinomas genome instability and sensitizes this cancer to therapeutics inducing DNA damaging agents [31]. Subsequently, inhibiting ATM activity sensitizes tumors to various therapeutic treatments.

We examined phosphorylated ATM, an early substrate exhibitive of ATM activity, in RYBP negative and RYBP positive U2OS cells. Cells were treated with 2 different DNA-damaging agents (radiomimetic and topoisomerase-I inhibitor) to determine whether RYBP inhibits ATM activity. Our results showed that cells overexpressing RYBP had fewer p-ATM foci than RYBP negative cells, revealing that RYBP reduces ATM autophosphorylation and ultimately, ATM activity at DNA damage sites. Interestingly, a previous study showed that ATM activation persists in cancer cells and mediates selective killing effects of the viral protein, apoptin [32]. Apoptin was shown to localize to the nucleus and induce cell death only when ATM is activated [32]. Upon inhibition of ATM kinase, apoptin is translocated to cytosol [32]. Although RYBP and apoptin were found to interact and exert selective tumor killing, overexpression of RYBP was shown to induce apoptin translocation back to cytosol [33]; similar to the effect of ATM inhibition [32]. These previous observations are consistent with our new findings that high levels of RYBP indeed inhibit ATM activity.

As earlier discussed, another bona fide substrate of ATM is checkpoint kinase, Chk2 [34]. The Chk2 is a component of the DDR that mediates cell cycle arrest and apoptosis [34]. When DNA damage occurs, ATM phosphorylates Chk2 at threonine 68 residue, and promotes oligomerization for Chk2 to auto-phosphorylate threonine 383 and 387 residues for full activation [34,35]. In our study, we measured the phosphorylated Chk2 and total Chk2 ratios in U2OS cells transfected with GFP-RYBP and those with GFP control after camptothecin-induced DNA damage. Our data showed no change in the total Chk2 across samples, but a distinct difference between the p-Chk2 and total Chk2 ratios. There was a prominent increase in the levels of p-Chk2/Chk2 ratios in control cells, compared to RYBP-expressing cells at 6 and 10 h after camptothecin treatments. However, these differences between groups at different time intervals were not statistically significant (when tested by two-way ANOVA). One reason for this may be attributed to Chk2 also being phosphorylated by ATR on threonine 68 residue, as well as ATM [1]. Overall, RYBP-expressing cells had reduced ATM activity compared to the control, as represented by lower p-Chk2/Chk2 ratios.

We further examined the effect of high RYBP levels on phosphorylated Chk2 levels in SKOV3 cells in the presence of DNA damage. High RYBP in SKOV3 reduced p-Chk2/Chk2 levels, compared to the GFP control cells at 1 h and 10 h after camptothecin treatment. Although this effect was more distinct in U2OS cells, the data from SKOV3 cells further supports that high RYBP levels reduce phosphorylation of Chk2 and thus, ATM activity. It was reported that ovarian cancer cells in general show higher ATM level/activity than normal cells [36]. Clinically, the higher the ATM level was, the more aggressive the ovarian cancer would be [36]. Therefore, differential ATM expression level/activity between osteosarcoma and ovarian cancer cells could explain the differences seen in our results. However, these results reinforce our findings that high levels of RYBP inhibit, or at least, reduce the phosphorylation of Chk2 (an ATM substrate), and therefore ATM activity.

Synergistic effects between ATM and PARP inhibitors were observed in previous studies [37,38,39]. Namely, PARP-1 has been shown to play a role in ATM activation as when PARP-1 is absent, ATM activation is deficient in the presence of DNA DSBs [37]. When PARP inhibitors were used, DNA DSBs were formed, thus activating ATM to mediate the DDR [37]. Subsequently, ATM deficient cells were shown to be more sensitive to cell death by PARP inhibition [37]. Moreover, Wang et al. (2017) found that Mantle cell lymphoma with reduced ATM expression was sensitized to olaparib in both in vitro and in vivo models [38]. Furthermore, Lloyd et al. (2020) examined the effect of olaparib in ATM deficient cell lines, with the addition of ATR inhibitor AZD6738, and found that this combination results in cell death within one to two cell divisions [39].

We also observed a similar synergy in U2OS cells when using both PARP and ATM inhibitors based on the 8.5-fold decrease of the IC_50_ for PARP inhibitor in the presence of ATM inhibitor. This synergy, however, was lost in RYBP-expressing cells with a less than 2-fold decrease in IC_50_, indicating that RYBP already inhibits, or at least in part, reduces ATM activity. This reduction in ATM activity as a result of overexpression of RYBP has also led to sensitization of RYBP-expressing U2OS cells to PARP inhibitors.

Xiang et al. reported that SKOV3 cells were more resistant to PARP inhibitors [27]. In fact, both Olaparib and AG14361 significantly inhibited the proliferation and invasion of A2780 cells but not SKOV3 cells [27]. Our results confirmed these observations as the IC_50_ of the PARP inhibitor (ABT-888) was shown to be the highest in SKOV3 (340 µM). Adding the ATM inhibitor, however, slightly increased the potency of the PARP inhibitor in control SKOV3 cells but not to the extent seen in U2OS. More importantly, the IC_50_ of the PARP inhibitor in RYBP-expressing SKOV3 did not change in the presence of the ATM inhibitor. This last observation paralleled our results in U2OS cells. We attribute the differences in the results between U2OS and SKOV3 cells to the higher resistance of SKOV3 to PARP inhibitors and to the higher ATM activity previously reported in ovarian cancer cells [27,36]. A recent study showed that inhibiting ATM activity by KU-60019 in SKOV3 resulted in cell apoptosis [40]. However, it was shown that KU-60019 is 10-fold more effective than KU-55933 (the ATM inhibitor used in our study) at blocking radiation-induced phosphorylation of ATM and its targets [41]. We, however, conclude that targeting ATM in SKOV3 is a potential important strategy in disrupting ovarian cancer proliferation, and overexpressing RYBP may offer new avenues for inhibiting ATM activity.

Additionally, a previous study examined the role of ATM in cancer cell migration and metastasis in breast cancer cells [28]. Chen et al. (2015) discovered that when ATM is inhibited, cell migration and invasion is dramatically reduced [28]. Interestingly, RYBP has been also shown to inhibit cancer cell migration and invasion in breast, lung and hepatocellular carcinomas [9,42,43]. As a result, we conducted wound closure assays on U2OS and MDA-MB-231 cells to determine whether overexpressing RYBP exhibited similar effects on cell migration, congruent with those of inhibiting ATM. With either overexpression of RYBP or inhibition of ATM, cells displayed similar inhibition of cell migration. On the other hand, inhibiting ATM activity in RYBP-expressing cells did not additionally reduce cell migration. This serves as another line of evidence to support our findings that RYBP does, in fact, inhibit cancer cell migration by, at least in part, inhibiting ATM activity. Likewise, this provides additional insights into how RYBP sensitizes cancer cells to DNA damaging agents, e.g., PARP inhibitors.

## 4. Materials and Methods

### 4.1. Antibodies and Reagents

The reagents and antibodies were purchased from the following sources, respectively: Dulbecco’s modified Eagle’s medium (DMEM), Corning; Leibovitz’s L15 (Modified) 1× w/L-glutamine; McCoy’s 5A Complete Medium; fetal bovine serum (FBS), Krackler; p-Chk2 (2197S), Chk2 (2662T), Chk2 (3440T), p-ATM (5883S), ATM (2873S), Cell Signalling, Danvers, MA, USA; RYBP (ab185971), GFP (ab290), Abcam, Cambridge, MA, USA; PARP inhibitor Veliparib (ABT-888); ATM inhibitor (KU-55933); MTT Cell Proliferation Kit, Ozbiosciences (MT01000); Effectene Transfection Reagent, Qiagen (301425).

### 4.2. Cell Culture

The human osteosarcoma cell line, U2OS, (HTB-96, ATCC) used in this study was cultured at 37 °C in a humidified 5% CO_2_ atmosphere in DMEM medium supplemented with 10% FBS. The human breast cancer cell line, MDA-MB-231, (HTB-26, ATCC) used in this study was cultured at 37 °C in a humidified 0% CO_2_ atmosphere in Leibovitz’s L15 (Modified) 1× w/L-glutamine supplemented with 10% FBS. The human ovarian cancer cell line, SKOV3 (HTB-77, ATCC), used in this study was cultured at 37 °C in a humidified 5% CO_2_ atmosphere in McCoy’s 5A Complete Medium supplemented with 10% FBS.

### 4.3. Western Blotting

U2OS and SKOV3 cells as described in the text were lysed in 500 µL RIPA buffer with phosphatase and protease inhibitor, quantified by the BCA protein assay, and 20 µg of total protein were separated by gel electrophoresis and transferred onto PVDF membranes. After the transfer, the membranes were blocked with 5% milk in TBS-T (50 mM Tris pH 8.4, 0.9% NaCl, 0.05% Tween-20) for 1 h and incubated overnight at 4 °C in the presence of the previously listed primary antibodies diluted in TBS-T (1:1000; 1:5000 dilutions). Membranes were washed with TBS-T and incubated with anti-rabbit (1:10,000 dilution) or anti-mouse (1:5000) secondary antibody for 1 h at 25 ℃ and washed with TBS-T before development and immunoreactive bands revelation by HRP/ECL reaction (ECL, Bio-Rad, Hercules, CA, USA). A total of *n* = 3 for cell treatment and Western blot were performed.

### 4.4. Immunofluorescence Microscopy

U2OS cells were grown on 18 × 18 mm poly-L-lysine coated coverslips for 24 h to reach 70% confluency. Cells were transfected with 1 µg of pGFP-RYBP-GW [2] using Qiagen Effectene Transfection Reagent (8 µL enhancer, 25 µL Effectene reagent, total 150 µL buffer) and incubated for additional 24 h. DNA damage was induced by adding calicheamicin (50 pM) or camptothecin (10 µM) for 1 h. Then, cells were fixed and permeabilized with chilled methanol for 10 min in −20 °C. Cells were washed with 0.1% Triton-X in PBS and PBS. Cells were incubated with anti-p-ATM primary antibody (1:500 dilution) for 1 h at 25 °C and washed with PBS before the incubation with anti-rabbit Alexa Fluor^®^ 568 secondary antibody for 1 h (1:200 dilution). The cells were washed one last time with PBS prior to mounting the coverslips with EMS Shield Mounting Medium with DAPI for nuclear staining and anti-fading agent, DABCO (EMS, 17989). The cells were imaged with a Leica laser scanning confocal microscope using a 40× oil-immersion objective lens.

### 4.5. Cell Survival Assay

In a 96-well plate, 5000 U2OS cells per well were seeded and allowed to attach overnight under standard culture conditions. The following day, concentrations of 2.5–100 μM of the PARP inhibitor ABT-888 and ±5.0 µM of ATM inhibitor KU-55933 were added, and cells were incubated with these compounds for 3 days. At the completion of the incubation, media with compounds was removed, the cells were washed with PBS and 100 µL of 1× MTT (3-(4,5-dimethylthiazol-2-yl)-2,4-diphenyltetrazolium bromide) working solution was added to each well for a 4-h incubation at 37 °C. After 4 h, MTT solubilization solution was added as per manufacturer’s instructions (MTT Cell Proliferation Kit, Ozbiosciences, MT01000). The absorbance was measured at 570 nm and 650 nm, background was removed, and absorbance was normalized to untreated controls to determine compound effects on cell viability. SKOV3 cells (PARP-resistant), 4000 cells per well seeded in a 96 well plate. The following day, concentrations of 30–1000 μM of the PARP inhibitor ABT-888 and ±30.0 μM of ATM inhibitor KU-55933 were added, and cells were incubated with these compounds for 3 days. The remaining protocol was followed as in U2OS cells.

### 4.6. Wound Closure Assay

In 6 well plates, U2OS or MDA-MB-231 cells were seeded and allowed to attach overnight under standard culture conditions. When the cells reached 70% confluence, four of the six wells were transfected with 1 µg of peGFP-RYBP-GW [2] using Qiagen Effectene Transfection Reagent (8 µL enhancer, 25 µL effectene reagent, total 150 µL buffer) at 70% confluency; the other two wells served as vehicle controls. Once the wells reached 100% confluency, a vertical wound/scratch was made using a 200 µL pipette tip. Cells were washed with PBS and Opti-MEM was replaced in each well. Three of the wells (one vehicle control and two transfected wells) were treated with 30 µM ATM inhibitor, KU-55933, for 72 h. Images were taken every 24 h of the same area of the wound in each well. The percent wound closure was calculated by final wound width subtracted from initial wound width (µm) divided by the initial wound width.

### 4.7. Statistical Analysis

All results are expressed as mean ± standard error of the mean. The significant differences between the mean values were analyzed by one-way or two-way ANOVA using GraphPad Prism-9 for immunoblotting, cell survival and wound closure assays and Student’s *t*-test for p-ATM foci in immunofluorescence data. Significant differences are reported as * *p* < 0.05.

The concentrations of PARP inhibitor which exhibited 50% cell viability (IC_50_) were calculated by nonlinear regression method using GraphPad Prism-9.

## 5. Conclusions

In conclusion, we showed that cancer cells expressing high levels of RYBP have lower ATM activity. This lower ATM activity was associated with sensitizing RYBP-expressing cells to PARP inhibitor and reducing cancer migration. Accordingly, we propose that higher expression of RYBP will be associated with better prognosis in cancer patients. Clinically, we predict that patients with high-RYBP cancers will respond better to PARP inhibitors. However, when ATM inhibitors are approved, we speculate that the reported synergy between PARP inhibitors and ATM inhibitors may be lost in high-RYBP tumors.

## Figures and Tables

**Figure 1 ijms-23-11764-f001:**
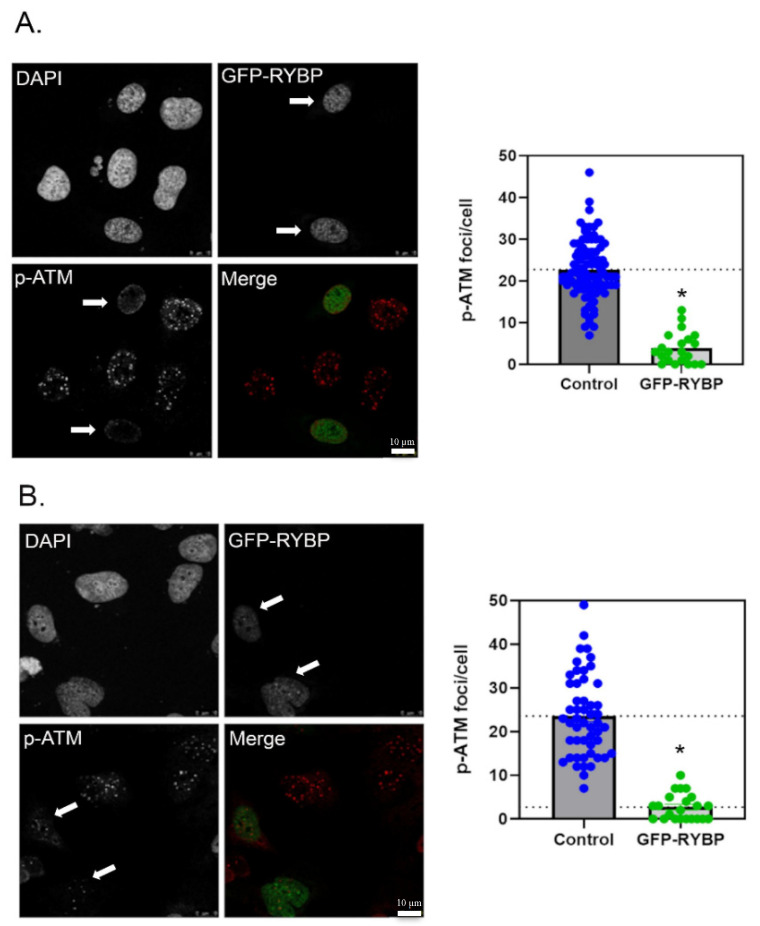
Effect of RYBP expression on phosphorylation of ATM after DNA damage. (**A**). Left; cells transfected with GFP-RYBP and treated with calicheamicin (50 pM, 1 h). Arrows indicate that cells expressing GFP-RYBP have less p-ATM foci. Right; quantification of p-ATM foci per nucleus (at least 50 nuclei from 3 independent experiments) in control and RYBP-expressing cells. (**B**). Left; cells transfected with GFP-RYBP and treated with camptothecin (10 µm, 1 h). Arrows indicate that cells expressing GFP-RYBP have less p-ATM foci. Right; quantification of p-ATM foci per nucleus (at least 50 nuclei from 3 independent experiments) in control and RYBP-expressing cells. * *p* < 0.5 (Student’s *t*-test). In the merged images, GFP-RYBP (green) and p-ATM (red). Scale bar is 10 μm.

**Figure 2 ijms-23-11764-f002:**
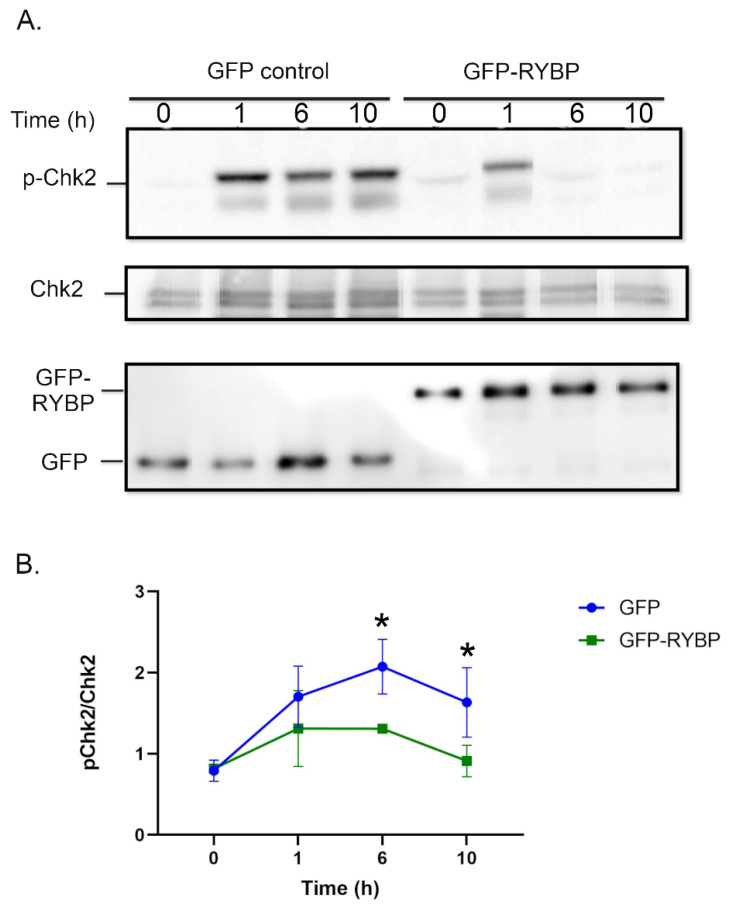
Effect of RYBP expression on phosphorylation of Chk2 after DNA damage. (**A**). Levels of p-Chk2 and total Chk2 are shown after camptothecin (1 µM) treatment at 0-, 1-, 6- and 10-h time intervals for GFP-control and GFP-RYBP expressing cells. (**B**). Quantification of p-Chk2/Chk2 ratios for GFP-control and GFP-RYBP expressing cells (*n* = 3 independent experiments). * *p* < 0.5 in comparison to 0-h time-point in GFP-control cells (one-way ANOVA).

**Figure 3 ijms-23-11764-f003:**
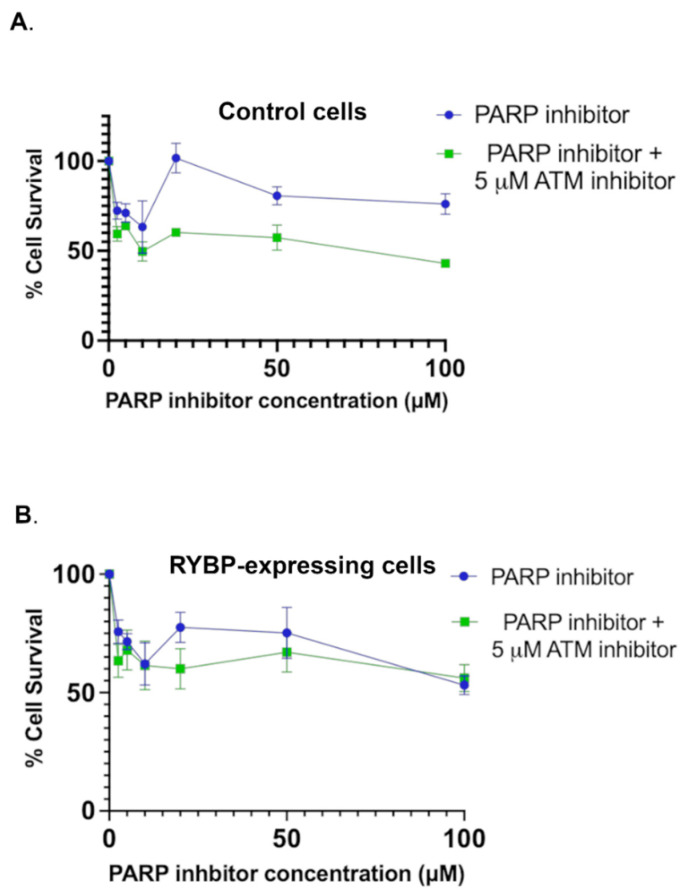
Effect of RYBP on the synergy between PARP inhibitor and ATM inhibitor. (**A**). In control U2OS cells, addition of ATM inhibitor (KU-55933) dramatically reduced the IC_50_ of PARP inhibitor (ABT-888) from 220 µM to 26 µM. (**B**). In RYBP-expressing cells, addition of ATM inhibitor (KU-55933) slightly reduced the IC_50_ of PARP inhibitor (ABT-888) from 83 µM.

**Figure 4 ijms-23-11764-f004:**
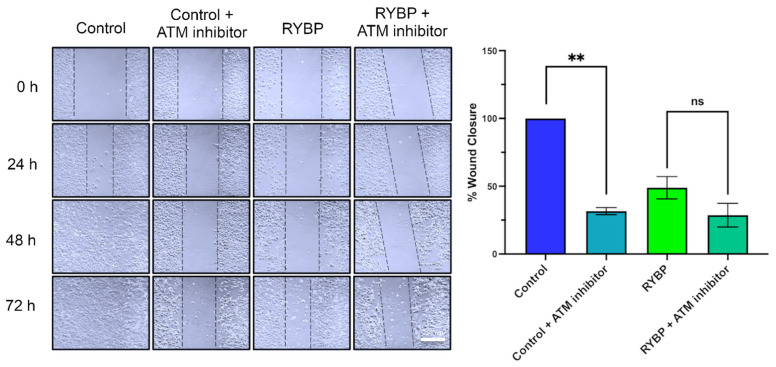
Effects of RYBP and ATM inhibition on U2OS cell migration. Left; wound closure assays of control cells ± ATM inhibitor and RYBP-expressing cells ± ATM inhibitor. Right; quantification of % wound closure at 72-h time-point. RYBP alone significantly reduced migration of U2OS cells and further addition of ATM inhibitor in RYBP-expressing cells did not further reduce cell migration. ** *p* < 0.1 in comparison to control (one-way ANOVA). ns; non-significant.

## Data Availability

Not applicable.

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
