# Peer review of "RYBP Sensitizes Cancer Cells to PARP Inhibitors by Regulating ATM Activity"

_ijms, 2022, doi:10.3390/ijms231911764_

Round 1

Reviewer 1 Report

1 - The WB in Figure 2a needs to be repeated in triplicate; the original uncropped images need to be provided in the SI. The authors must note that the GFP-RYBP expression is not equal among the four lanes. 

2 - The experiment in Figure 3 needs a control where the authors use ATM inhibitor alone. Then the authors need to use the Compusyn software to calculate synergy.

3 - The authors need to repeat the experiments shown in Figures 2 and 3 using a second cell line beside U2OS cells

4 - The authors mentioned in the introduction that "albeit ATM is no longer interconnected to cell-cycle arrest and apoptosis," While the first sentence of the discussion section claims that ATM is interconnected to cell-cycle arrest.

5 - The title must specify the DNA damage agents used; it is too broad.

Reviewer 2 Report

A manuscript entitled " RYBP sensitizes cancer cells to DNA damaging agents by regulating ATM activity" provides RYBP inhibits ATM activity and RYBP expression sensitizes cancer cells to DNA damaging agents. 

1. In the abstract, what is the full name of BRCA and PARP? Please add.

2. In the figure 2, for the camptothecin treated, why the author choose 1, 6, and 10 hours to induce DNA damage rather than continuous time?

Round 2

Reviewer 1 Report

The authors need to publish all the uncropped WB images provided as part of the supporting information.